# Towards Better Generalization of Adaptive Gradient Methods

**Yingxue Zhou, Belhal Karimi, Jinxing Yu, Zhiqiang Xu, Ping Li**
Cognitive Computing Lab
Baidu Research
No.10 Xibeiwang East Road, Beijing 100193, China
10900 NE 8th St. Bellevue, Washington 98004, USA
{hustzhouyx, belhal.karimi, jinxingyu18, zhiqiangxu2001, pingli98}@gmail.com

## Abstract

Adaptive gradient methods such as AdaGrad, RMSprop and Adam have been optimizers of choice for deep learning due to their fast training speed. However, it was recently observed that their generalization performance is often worse than that of SGD for over-parameterized neural networks. While new algorithms (such as AdaBound) have been proposed to improve the situation, the provided analyses are only committed to optimization bounds for the training objective, leaving critical generalization capacity unexplored. To close this gap, we propose *Stable Adaptive Gradient Descent* (SAGD) for non-convex optimization which leverages differential privacy to boost the generalization performance of adaptive gradient methods. Theoretical analyses show that SAGD has high-probability convergence to a population stationary point. We further conduct experiments on various popular deep learning tasks and models. Experimental results illustrate that SAGD is empirically competitive and often better than baselines.

## 1   Introduction

In this work, we consider the stochastic non-convex optimization [40] problem which approximately minimizes the *population loss* given $n$ i.i.d. samples $\mathbf{z}_1, \ldots, \mathbf{z}_n$. Mathematically speaking, we consider the following optimization problem:

$$\min_{\mathbf{w} \in \mathcal{W}} \ f(\mathbf{w}) \triangleq \mathbb{E}_{\mathbf{z} \sim \mathcal{P}}[\ell(\mathbf{w}, \mathbf{z})] \,, \tag{1}$$

where $\mathbf{z} \in \mathcal{Z}$ is a data sample in domain $\mathcal{Z}$ following an unknown sample distribution $\mathcal{P}$, $\mathbf{w}$ represents the parameter of the underlying learning model, $\ell : \mathcal{W} \times \mathcal{Z} \mapsto \mathbb{R}$ is a certain loss function associated with the learning problem, and the loss function $f$ defined by the population risk is non-convex as with most deep learning tasks. Since finding the global minimum for non-convex functions is NP-hard, the utility of a parameter is usually measured by the $\ell_2$-norm of the gradient.

Due to the unavailability of distribution $\mathcal{P}$, the challenge of a learning algorithm is to search for an approximate minimizer of $f(\mathbf{w})$ based on only $n$ samples $\mathbf{z}_1, \ldots, \mathbf{z}_n$. A natural approach toward solving the problem stated in (1) is empirical risk minimization (ERM) [33], which minimizes the empirical risk: $\min_{\mathbf{w} \in \mathcal{W}} \hat{f}(\mathbf{w}) \triangleq \frac{1}{n} \sum_{j=1}^{n} \ell(\mathbf{w}, \mathbf{z}_j)$, where $\hat{f}(\mathbf{w})$ is referred to as empirical risk.

Stochastic gradient descent (SGD) [32] which iteratively updates the parameter of a model by descending in the direction of the negative gradient, computed on a single sample or a mini-batch of samples, has been the most dominant algorithm for solving the ERM problem, e.g., training deep neural networks. Since the learning rate has a crucial impact on the convergence and performance of SGD algorithms, there have been studies (e.g., [5]) which automatically tune the learning rate by

reducing it each time a stationarity is detected. A different (and popular) strategy for automatically tuning the learning-rate decay is to use adaptive gradient methods, such as AdaGrad [10], RM-Sprop [35], and Adam [19], which have emerged leveraging the curvature of the objective function resulting in adaptive coordinate-wise learning rates for faster convergence.

However, the generalization ability of these adaptive methods is often worse than that of SGD for over-parameterized neural networks, e.g., convolutional neural network (CNN) for image classification and recurrent neural network (RNN) for language modeling [39]. To mitigate this issue, several recent algorithms were proposed to combine adaptive methods with SGD. For example, AdaBound [24] and SWAT [18] switch from Adam to SGD as the training proceeds, while Padam [6, 41] unifies AMSGrad [31] and SGD with a partially adaptive parameter. Despite much efforts on deriving theoretical convergence results of the objective function [40, 38, 43, 8], these newly proposed adaptive gradient methods are often misunderstood regarding their generalization abilities, which is the ultimate goal. On the other hand, current adaptive gradient methods [10, 19, 35, 31, 38, 7] follow a typical stochastic optimization (SO) oracle paradigm [32, 15] which uses stochastic gradients to update the parameters. The SO oracle requires *new samples* at every iteration to get the stochastic gradient such that, in expectation, it equals the *population* gradient. In practice, however, only finite training samples are available and reused by the optimization oracle for a certain number of times (i.e., epochs). [16] found that the generalization error increases with the number of times the optimization oracle passes over the training data. It is thus expected that gradient descent algorithms will be much more well-behaved if we have access to an infinite number of fresh samples. Re-using data samples is therefore a caveat for the generalization of a given algorithm.

To tackle the above issues, we propose *Stable Adaptive Gradient Descent* (SAGD) which aims at improving the generalization of general adaptive gradient descent algorithms. SAGD behaves similarly to the aforementioned ideal case of infinite fresh samples borrowing ideas from *adaptive data analysis* [11] and *differential privacy* [14]. The main idea of our method is that, at each iteration, SAGD accesses the training set through a differentially private mechanism and computes an estimated gradient of the objective function $\nabla f(\mathbf{w})$. It then uses the estimated gradient to perform a descent step with adaptive stepsize. We prove that the reused data points in SAGD nearly possess the statistical nature of *fresh samples* yielding to high concentration bounds of the population gradients through the iterations. **Our contributions** can be summarized as follows:

- We derive a novel adaptive gradient method, namely SAGD, leveraging ideas of differential privacy and adaptive data analysis aiming at improving the generalization of current baseline methods. A mini-batch variant is also introduced for large-scale learning tasks.

- Our differentially private mechanism, embedded in the SAGD, explores the idea of Laplace Mechanism (adding Laplace noises to gradients) and SPARSE VECTOR TECHNIQUE [14] leading to DPG-LAP and DPG-SPARSE methods saving privacy cost. In particular, we show that differentially private gradients stay close to the population gradients with high probability.

- We establish various theoretical guarantees for our algorithm. We derive a concentration bound on the generalization error and show that the $\ell_2$-norm of the *population gradient*, i.e., $\|\nabla f(\mathbf{w})\|$ obtained by the SAGD converges in $\tilde{\mathcal{O}}(1/n^{2/3})$ with high probability.

- We conduct several experimental applications based on training neural networks for image classification and language modeling indicating that SAGD outperforms existing adaptive gradient methods in terms of the generalization and over-fitting performance.

**Roadmap:** The SAGD algorithm, including the differentially private mechanisms, and its mini-batch variant are described in Section 3. Numerical experiments are presented Section 4. Section 5 concludes our work. Due to space limit, most of the proofs are relegated to supplementary material.

**Notations:** We use $\mathbf{g}_t$ and $\nabla f(\mathbf{w})$ interchangeably to denote the *population gradient* such that $\mathbf{g}_t = \nabla f(\mathbf{w}_t) = \mathbb{E}_{\mathbf{z} \in \mathcal{P}}[\nabla \ell(\mathbf{w}_t, \mathbf{z})]$. $S = \{\mathbf{z}_1, \ldots, \mathbf{z}_n\}$ denotes the $n$ available training samples. $\hat{\mathbf{g}}_t$ denotes the sample gradient evaluated on $S$ such that $\hat{\mathbf{g}}_t = \nabla \hat{f}(\mathbf{w}) = \frac{1}{n} \sum_{j=1}^{n} \nabla \ell(\mathbf{w}_t, \mathbf{z}_j)$. For a vector $\mathbf{v}$, $\mathbf{v}^2$ represents that $\mathbf{v}$ is element-wise squared. We use $\mathbf{v}^i$ or $[\mathbf{v}]_i$ to denote the $i$-th coordinate of $\mathbf{v}$. We use $\|\mathbf{v}\|_2$ and $\|\mathbf{v}\|$ alternatively to denote the $\ell_2$-norm of $\mathbf{v}$. We write $\|\mathbf{v}\|_1$ as the $\ell_1$-norm of $\mathbf{v}$ and denote $[d] = \{1, \ldots, d\}$.

## 2 Preliminaries

**Adaptive Gradient Methods:** In the non-convex setting, existing work on SGD [15] and adaptive gradient methods [40, 38, 43, 8] show convergence to a stationary point with a rate of $\mathcal{O}(1/\sqrt{T})$ where $T$ is the number of stochastic gradient computations. Given $n$ samples, a stochastic oracle can obtain at most $n$ stochastic gradients, which implies convergence to the population stationarity with a rate of $\mathcal{O}(1/\sqrt{n})$. In addition, [21, 30, 16, 27, 28, 8, 23] study the generalization of gradient-based optimization algorithms using the generalization property of stable algorithms [2]. In particular, [30, 27, 23, 28] focus on noisy gradient algorithms, e.g., SGLD, and provide a generalization error (population risk minus empirical risk) bound in $\mathcal{O}(\sqrt{T}/n)$. This type of bounds usually has a dependence on the training data and has a polynomial dependence on $T$.

**Differential Privacy and Adaptive Data Analysis:** Differential privacy [14] was originally studied for preserving the privacy of individual data in the statistical query. Recently, differential privacy has been widely used for stochastic optimization. Some pioneering work [4, 1, 37] introduce differential privacy to empirical risk minimization (ERM) to protect sensitive information of the training data. The popular differentially private algorithms include the gradient perturbation that adds noise to the gradient in gradient descent algorithms [4, 1, 36]. Moreover, in Adaptive Data Analysis ADA [11, 12, 13], the same holdout set is used multiple times to test the hypotheses which are generated based on previous test results. It has been shown that reusing the holdout set via a differentially private mechanism ensures the validity of the holdout set. In other words, the differentially private reused dataset maintains the statistical nature of fresh samples and improves generalization [42].

## 3 Stable Adaptive Gradient Descent Algorithm

Beforehand, we recall the definition of an $(\epsilon, \delta)$-differentially private algorithm:

**Definition 1.** *(Differential Privacy [14]) A randomized algorithm $\mathcal{M}$ is $(\epsilon, \delta)$-differentially private if*

$$\mathbb{P}\{\mathcal{M}(\mathcal{D}) \in \mathcal{Y}\} \leq \exp(\epsilon)\mathbb{P}\{\mathcal{M}(\mathcal{D}\prime) \in \mathcal{Y}\} + \delta$$

*holds for all $\mathcal{Y} \subseteq Range(\mathcal{M})$ and all pairs of adjacent datasets $\mathcal{D}, \mathcal{D}\prime$ that differ on a single sample.*

Intuitively, differential privacy means that the outcomes of two nearly identical datasets should be nearly identical such that an analyst will not be able to distinguish any single data point by monitoring the change of the output. One of the general approaches for achieving $(\epsilon, \delta)$-differential privacy when estimating a deterministic real-valued function $q : \mathcal{Z}^n \to \mathbb{R}^d$ is Laplace Mechanism [14], which adds Laplace noise calibrated to the function $q$, i.e., $\mathcal{M}(\mathcal{D}) = q(\mathcal{D}) + \mathbf{b}$, where for all $i \in [d]$, $\mathbf{b}^i \sim \text{Laplace}(0, \sigma^2)$. The value of $\sigma$ is decided by the privacy parameter $\epsilon$ and $\delta$. We present SAGD with two different **D**ifferential **P**rivate **G**radient (DPG) computing methods that provide an estimate of the gradient $\nabla f(\mathbf{w})$, namely DPG-LAP based on the *Laplace Mechanism* [14], see Section 3.1 and an improvement named DPG-SPARSE motivated by sparse vector technique [14] in Section 3.2.

### 3.1 SAGD with DGP-LAP

In most deep learning applications, a training set $S$ of size $n$ is observed. Then, at each iteration $t \in [T]$, SAGD, described in Algorithm 1, calls DPG-LAP (Line 5 in Algorithm 1), that computes

---

**Algorithm 1** SAGD with DGP-LAP

---

1: **Input**: Dataset $S$, certain loss $\ell(\cdot)$, initial point $\mathbf{w}_0$ and noise level $\sigma$.
2: Set noise level $\sigma$, iteration number $T$, and stepsize $\eta_t$.
3: **for** $t = 0, ..., T - 1$ **do**
4:      DPG-LAP: Compute full batch gradient on $S$:
$$\hat{\mathbf{g}}_t = \frac{1}{n} \sum_{j=1}^n \nabla \ell(\mathbf{w}_t, z_j).$$
5:      Set $\tilde{\mathbf{g}}_t = \hat{\mathbf{g}}_t + \mathbf{b}_t$, where $\mathbf{b}_t^i$ is drawn i.i.d from $\text{Lap}(\sigma)$ for all $i \in [d]$.
6:      $\mathbf{m}_t = \tilde{\mathbf{g}}_t$ and $\mathbf{v}_t = (1 - \beta_2) \sum_{i=1}^t \beta_2^{t-i} \tilde{\mathbf{g}}_i^2$.
7:      $\mathbf{w}_{t+1} = \mathbf{w}_t - \eta_t \mathbf{m}_t / (\sqrt{\mathbf{v}_t} + \nu)$.
8: **end for**

---

the empirical gradient noted $\tilde{\mathbf{g}}_t$ and updates the model parameter $\mathbf{w}_{t+1}$ using adaptive stepsize. Note that the noise variance $\sigma^2$, step-size $\eta_t$, iteration number $T$, $\beta_2$ are parameters and play an important role for our theoretical study presented in the sequel. We first consider DPG-LAP which adds Laplace noise $\mathbf{b}_t \in \mathbb{R}^d$ to the empirical gradient $\hat{\mathbf{g}}_t = \frac{1}{n} \sum_{j=1}^{n} \nabla \ell(\mathbf{w}_t, \mathbf{z}_j)$ and returns a noisy gradient $\tilde{\mathbf{g}}_t = \hat{\mathbf{g}}_t + \mathbf{b}_t$ to the optimization oracle Algorithm 1. Throughout this paper, we assume:

**A1.** *The objective function $f : \mathbb{R}^d \to \mathbb{R}$ is bounded from below by $f^\star$ and is L-smooth (L-Lipschitz gradients), i.e., $\|\nabla f(\mathbf{w}) - \nabla f(\mathbf{w}')\| \le L\|\mathbf{w} - \mathbf{w}'\|$, for all $\mathbf{w}, \mathbf{w}' \in \mathcal{W}$.*

**A2.** *The gradient of $\ell$ and its noisy approximation are bounded: For all $\mathbf{w} \in \mathcal{W}$, $\mathbf{z} \in \mathcal{Z}$ $\|\nabla \ell(\mathbf{w}, \mathbf{z})\| \le G$, for all $t \in [T]$, $\|\tilde{\mathbf{g}}_t\| \le G$, and $\|\nabla \ell(\mathbf{w}, z)\|_1 \le G_1$.*

To analyze the convergence of SAGD in terms of $\ell_2$ norm of the population gradient, we need to show that $\tilde{\mathbf{g}}_t$ approximate the population gradient $\mathbf{g}_t$ with high probability, i.e., the estimation error $\|\tilde{\mathbf{g}}_t - \mathbf{g}_t\|$ is small at every iteration. To make such an analysis, we first present the generalization guarantee of any differentially private algorithm in Lemma 1, and then show that SAGD is differentially private in Lemma 2. It is followed by establishing SAGD's generalization guarantee in Theorem 1, i.e., estimated $\tilde{\mathbf{g}}_t$ approximates the population gradient $\mathbf{g}_t$ with high probability.

*High-probability bound:* We first show that the noisy gradient $\tilde{\mathbf{g}}_t$ approximates the population gradient $\mathbf{g}_t$ with high probability. A general approach for analyzing the estimation error of sample gradient to population gradient is Hoeffding's bound, i.e., given training set $S \in \mathcal{Z}^n$, and a fixed $\mathbf{w}_0$ chosen to be independent of the dataset $S$, denote the empirical gradient $\hat{\mathbf{g}}_0 = \mathbb{E}_{z \in S} \nabla \ell(\mathbf{w}_0, z)$ and population gradient $\mathbf{g}_0 = \mathbb{E}_{z \sim \mathcal{P}}[\nabla l(\mathbf{w}_0, z)]$, Hoeffding's bound implies for $i \in [d]$ and $\mu > 0$:

$$P\{|\hat{\mathbf{g}}_0^i - \mathbf{g}_0^i| \ge \mu\} \le 2 \exp\left(\frac{-2n\mu^2}{4G^2}\right) . \tag{2}$$

Generally, if $\mathbf{w}_1$ is updated using the gradient computed on training set $S$, i.e., $\mathbf{w}_1 = \mathbf{w}_0 - \eta \hat{\mathbf{g}}_0$, concentration inequality (2) *will not* hold for $\hat{\mathbf{g}}_1 = \mathbb{E}_{z \in S} \nabla_i \ell(\mathbf{w}_1, z)$, because $\mathbf{w}_1$ is no longer independent of $S$. However, Lemma 1 shows that if $\mathbf{w}_t, \forall t \in [T]$ is generated by reusing $S$ under a differentially private mechanism, concentration bounds similar to Eq. (2) will hold for all $\mathbf{w}_1, \mathbf{w}_2, ..., \mathbf{w}_T$ that are adaptively generated on the same dataset $S$.

**Lemma 1.** *Let $\mathcal{A}$ be an $(\epsilon, \delta)$-differentially private gradient descent algorithm with access to training set $S$ of size $n$. Let $\mathbf{w}_t = \mathcal{A}(S)$ be the parameter generated at iteration $t \in [T]$ and $\hat{\mathbf{g}}_t$ the empirical gradient on $S$. For any $\sigma > 0$, $\beta > 0$, if the privacy cost of $\mathcal{A}$ satisfies $\epsilon \le \sigma/13$, $\delta \le \sigma\beta/(26\ln(26/\sigma))$, and sample size $n \ge 2\ln(8/\delta)/\epsilon^2$, we then have*

$$\mathbb{P}\left\{|\hat{\mathbf{g}}_t^i - \mathbf{g}_t^i| \ge G\sigma\right\} \le \beta , \quad \forall i \in [d] \text{ and } \forall t \in [T] .$$

Lemma 1 is an instance of Theorem 8 from [11] and illustrates that, if the privacy cost $\epsilon$ is bounded by the estimation error, the differential privacy mechanism enables the reused training sample set to maintain statistical guarantees as if they were fresh samples. Then, we establish in Lemma 2, that SAGD with DPG-LAP is a differentially private algorithm with the following privacy cost:

**Lemma 2.** SAGD *with* DPG-LAP *(Alg. 1) is $(\frac{\sqrt{T \ln(1/\delta)} G_1}{n\sigma}, \delta)$-differentially private.*

In order to achieve a gradient concentration bound for SAGD with DPG-LAP as described in Lemma 1, we set $\sqrt{T \ln(1/\delta)} G_1/(n\sigma) \le \sigma/13$, $\delta \le \sigma\beta/(26\ln(26/\sigma))$, and sample size $n \ge 2\ln(8/\delta)/\epsilon^2$. Then, the following result shows that across all iterations, gradients produced by SAGD with DPG-LAP maintain high probability concentration bounds.

---

**Theorem 1** *Given $\sigma > 0$, let $\tilde{\mathbf{g}}_1, ..., \tilde{\mathbf{g}}_T$ be gradients computed by* DPG-LAP *in* SAGD. *Set the number of iterations $2n\sigma^2/G_1^2 \le T \le n^2\sigma^4/(169\ln(1/(\sigma\beta))G_1^2)$, then for $t \in [T]$, $\beta > 0$, $\mu > 0$:*

$$\mathbb{P}\left\{\|\tilde{\mathbf{g}}_t - \mathbf{g}_t\| \ge \sqrt{d}\sigma(G + \mu)\right\} \le d\beta + d\exp(-\mu) .$$

---

Note that given the concentration error bound of $\sqrt{d}\sigma(G + \mu)$, Theorem 1 indicates that a higher noise level $\sigma$, implying a better privacy guarantee and a larger number of iterations $T$, would meanwhile incur a larger concentration error. Thus, there is a trade-off between noise and accuracy illustrated by the positive numbers $\beta$ and $\mu$. A larger $\mu$ brings a larger concentration error but a smaller

probability. A larger $\beta$ implies a larger upper bound on $T$, yet also a larger probability bound. We optimize the choice of $\beta$ and $\mu$ for analyzing the convergence to the population stationary point.

*Non-asymptotic convergence rate:* We derive the optimal values of $\sigma$ and $T$ to improve the trade-off between the statistical rate and the optimization rate and we obtain a novel finite-time bound in Theorem 2. Denote $\rho_{n,d} \triangleq \mathcal{O}\left(\ln n + \ln d\right)$, we prove that SAGD with DPG-LAP converges to a population stationary point with high probability at the following rate:

---

**Theorem 2** *Given training set $S$ of size $n$, for $\nu > 0$, if $\eta_t = \eta$ with $\eta \leq \nu/(2L)$, $\sigma = 1/n^{1/3}$, iteration number $T = n^{2/3}/\left(169G_1^2(\ln d + 7\ln n/3)\right)$, $\mu = \ln(1/\beta)$ and $\beta = 1/(dn^{5/3})$, then* SAGD *with* DPG-LAP *algorithm yields:*

$$\min_{1 \leq t \leq T} \|\nabla f(\mathbf{w}_t)\|^2 \leq \mathcal{O}\left(\frac{\rho_{n,d}\left(f(\mathbf{w}_1) - f^\star\right)}{n^{2/3}}\right) + \mathcal{O}\left(\frac{d\rho_{n,d}^2}{n^{2/3}}\right),$$

*with probability at least $1 - \mathcal{O}\left(1/(\rho_{n,d}n)\right)$.*

---

Theorem 2 shows that, given $n$ samples, SAGD converges to a stationary point at a rate of $\tilde{\mathcal{O}}(1/n^{2/3})$ where we use the $\ell_2$ norm of the gradient of the objective function as a convergence criterion. Particularly, the first term of the bound corresponds to the optimization error $\mathcal{O}(1/T)$ with $T = \mathcal{O}(n^{2/3})$, while the second is the statistical error depending on available sample size $n$ and dimension $d$. The current optimization analyses [40, 38, 43, 8] show that adaptive gradient descent algorithms converge to the stationary point of the objective function with a rate of $\mathcal{O}(1/\sqrt{T})$ with $T$ stochastic gradient computations. Given $n$ samples, their analyses yield a rate of $\mathcal{O}(1/\sqrt{n})$. Thus, the SAGD achieves a sharper bound compared to the previous analyses.

## 3.2 SAGD with DPG-SPARSE

In this section, we consider the SAGD with an advanced version of DPG named DPG-SPARSE motivated by the sparse vector technique [14] aiming to provide a sharper result on the privacy cost $\epsilon$ and $\delta$. Lemma 2 shows that the privacy cost of SAGD with DPG-LAP scales with $\mathcal{O}(\sqrt{T})$. In order to guarantee the generalization of SAGD as stated in Theorem 1, we need to control the privacy cost below a certain threshold i.e., $\sqrt{T\ln(1/\delta)}G_1/(n\sigma) \leq \sigma/13$. However, it limits the iteration number $T$ of SAGD, leading to a compromised optimization term in Theorem 2. In order to relax the upper bound on $T$, we propose the SAGD with DPG-SPARSE in Algorithm 2. Given $n$ samples, Algorithm 2 splits the dataset evenly into two parts $S_1$ and $S_2$. At each iteration $t$, Algorithm 2 computes gradients on both datasets: $\hat{\mathbf{g}}_{S_1,t} = \frac{1}{|S_1|}\sum_{\mathbf{z}_j \in S_1}\nabla\ell(\mathbf{w}_t, \mathbf{z}_j)$ and $\hat{\mathbf{g}}_{S_2,t} = \frac{1}{|S_2|}\sum_{\mathbf{z}_j \in S_2}\nabla\ell(\mathbf{w}_t, \mathbf{z}_j)$. It then validates $\hat{\mathbf{g}}_{S_1,t}$ with $\hat{\mathbf{g}}_{S_2,t}$, i.e., if the norm of their difference is greater than a random threshold $\tau - \gamma$, it returns $\tilde{\mathbf{g}}_t = \hat{\mathbf{g}}_{S_1,t} + \mathbf{b}_t$, otherwise $\tilde{\mathbf{g}}_t = \hat{\mathbf{g}}_{S_2,t}$.

---

**Algorithm 2** SAGD with DPG-SPARSE

1: **Input**: Dataset $S$, certain loss $\ell(\cdot)$, initial point $\mathbf{w}_0$.
2: Set noise level $\sigma$, iteration number $T$, and stepsize $\eta_t$.
3: Split $S$ randomly into $S_1$ and $S_2$.
4: **for** $t = 0, ..., T-1$ **do**
5:     DPG-SPARSE: Compute full batch gradient on $S_1$ and $S_2$:
        $\hat{\mathbf{g}}_{S_1,t} = \frac{1}{|S_1|}\sum_{\mathbf{z}_j \in S_1}\nabla\ell(\mathbf{w}_t, \mathbf{z}_j), \quad \hat{\mathbf{g}}_{S_2,t} = \frac{1}{|S_2|}\sum_{\mathbf{z}_j \in S_2}\nabla\ell(\mathbf{w}_t, \mathbf{z}_j).$
6:     Sample $\gamma \sim \text{Lap}(2\sigma)$, $\tau \sim \text{Lap}(4\sigma)$.
7:     **if** $\|\hat{\mathbf{g}}_{S_1,t} - \hat{\mathbf{g}}_{S_2,t}\| + \gamma > \tau$ **then**
8:         $\tilde{\mathbf{g}}_t = \hat{\mathbf{g}}_{S_1,t} + \mathbf{b}_t$, where $\mathbf{b}_t^i$ is drawn i.i.d from $\text{Lap}(\sigma)$, for all $i \in [d]$.
9:     **else**
10:         $\tilde{\mathbf{g}}_t = \hat{\mathbf{g}}_{S_2,t}$
11:     **end if**
12:     $\mathbf{m}_t = \tilde{\mathbf{g}}_t$ and $\mathbf{v}_t = (1 - \beta_2)\sum_{i=1}^{t}\beta_2^{t-i}\tilde{\mathbf{g}}_i^2$.
13:     $\mathbf{w}_{t+1} = \mathbf{w}_t - \eta_t\mathbf{m}_t/(\sqrt{\mathbf{v}_t} + \nu)$.
14: **end for**
15: **Return**: $\tilde{\mathbf{g}}_t$.

---

Following THRESHOLDOUT, a sparse vector technique for adaptive data analysis, [42] propose a stable gradient descent algorithm which uses a similar framework as DPG-SPARSE to compute an estimated gradient by validating each coordinate of $\hat{\mathbf{g}}_{S_1,t}$ and $\hat{\mathbf{g}}_{S_2,t}$. Thus, their method is computationally expensive in high-dimensional settings such as deep neural networks.

*High-probability bound:* To analyze the privacy cost of DPG-SPARSE, let $C_s$ be the number of times the validation fails, i.e., $\|\hat{\mathbf{g}}_{S_1,t} - \hat{\mathbf{g}}_{S_2,t}\| + \gamma > \tau$ is true, over $T$ iterations in SAGD. The following Lemma establishes the privacy cost of the SAGD with DPG-SPARSE algorithm.

**Lemma 3.** SAGD *with* DPG-SPARSE *(Alg. 2) is* $(\frac{\sqrt{C_s \ln(2/\delta)}2G_1}{n\sigma}, \delta)$-*differentially private.*

Lemma 3 shows that the privacy cost of SAGD with DPG-SPARSE scales with $\mathcal{O}(\sqrt{C_s})$ where $C_s \leq T$. In other words, DPG-SPARSE procedure improves the privacy cost of the SAGD algorithm. Indeed, in order to achieve the generalization guarantee of SAGD with DPG-SPARSE, stated in Lemma 1 and by considering the result of Lemma 3, we only need to set $\sqrt{C_s \ln(1/\delta)}G_1/(n\sigma) \leq \sigma/13$, which potentially improves the upper bound on $T$. We derive the generalization guarantee of $\tilde{\mathbf{g}}_t$ generated by the SAGD with DPG-SPARSE algorithm in the following result:

> **Theorem 3** *Given* $\sigma > 0$, *let* $\tilde{\mathbf{g}}_1, ..., \tilde{\mathbf{g}}_T$ *be the gradients computed by* DPG-SPARSE *in* SAGD. *With a budget* $n\sigma^2/(2G_1^2) \leq C_s \leq n^2\sigma^4/(676\ln(1/(\sigma\beta))G_1^2)$, *then for* $t \in [T], \beta > 0, \mu > 0$:
> $$\mathbb{P}\left\{\|\tilde{\mathbf{g}}_t - \mathbf{g}_t\| \geq \sqrt{d}\sigma(1 + \mu)\right\} \leq d\beta + d\exp(-\mu).$$

In the worst case $C_s = T$, we recover the bound of $T \leq n^2\sigma^4/(676\ln(1/(\sigma\beta))G_1^2)$ of DPG-LAP.

*Non-asymptotic convergence rate:* The finite-time upper bound on the convergence criterion of interest for the SAGD with DPG-SPARSE algorithm (Algorithm 2) is stated as follows:

> **Theorem 4** *Given training set* $S$ *of size* $n$, *for* $\nu > 0$, *if* $\eta_t = \eta$ *which are chosen with* $\eta \leq \nu/(2L)$, *noise level* $\sigma = 1/n^{1/3}$, *and iteration number* $T = n^{2/3}/\left(676G_1^2(\ln d + \frac{7}{3}\ln n)\right)$, *then* SAGD *with* DPG-SPARSE *algorithm yields:*
> $$\min_{1 \leq t \leq T}\|\nabla f(\mathbf{w}_t)\|^2 \leq \mathcal{O}\left(\frac{\rho_{n,d}(f(\mathbf{w}_1) - f^\star)}{n^{2/3}}\right) + \mathcal{O}\left(\frac{d\rho_{n,d}^2}{n^{2/3}}\right),$$
> *with probability at least* $1 - \mathcal{O}\left(1/(\rho_{n,d}n)\right)$.

Theorem 4 displays a similar rate of $\mathcal{O}(1/n^{2/3})$ for the SAGD with DGP-SPARSE as Theorem 2. A sharper bound can be achieved when the number of validation failures $C_s$ is smaller than $T$. For example, if $C_s = \mathcal{O}(\sqrt{T})$, the upper bound of $T$ can be improved from $T \leq \mathcal{O}(n^2)$ to $T \leq \mathcal{O}(n^4)$.

### 3.3 Mini-batch Stable Adaptive Gradient Descent Algorithm

For large-scale learning we derive the mini-batch variant of SAGD in Algorithm 3. The training set $S$ is first partitioned into $B$ batches with $m$ samples for each batch. At each iteration $t$, Algorithm 3 uses any DPG procedure to compute a differential private gradient $\tilde{\mathbf{g}}_t$ on each batch and updates $\mathbf{w}_t$.

---

**Algorithm 3** Mini-Batch SAGD

---

1: **Input**: Dataset $S$, certain loss $\ell(\cdot)$, initial point $\mathbf{w}_0$.
2: Set noise level $\sigma$, epoch number $T$, batch size $m$, and stepsize $\eta_t$.
3: Split $S$ into $B = n/m$ batches: $\{s_1, ..., s_B\}$.
4: **for** $epoch = 1, ..., T$ **do**
5:     **for** $k = 1, ..., B$ **do**
6:         Call DPG-LAP or DPG-SPARSE to compute $\tilde{\mathbf{g}}_t$.
7:         $\mathbf{m}_t = \tilde{\mathbf{g}}_t$ and $\mathbf{v}_t = (1 - \beta_2)\sum_{i=1}^{t}\beta_2^{t-i}\tilde{\mathbf{g}}_i^2$.
8:         $\mathbf{w}_{t+1} = \mathbf{w}_t - \eta_t\mathbf{m}_t/(\sqrt{\mathbf{v}_t} + \nu)$.
9:     **end for**
10: **end for**

---

Theorem 5 describes the convergence rate of the mini-batch SAGD algorithm in terms of batch size $m$ and sample size $n$, i.e., $\mathcal{O}(1/(mn)^{1/3})$.

**Theorem 5** *Consider the mini-batch SAGD with DPG-LAP. Given $S$ of size $n$, with $\nu > 0$, $\eta_t = \eta \leq \nu/(2L)$, noise level $\sigma = 1/(mn)^{1/6}$, and epoch $T = m^{4/3}/\left(n^{2/3}169G_1^2(\ln d + \frac{7}{3}\ln n)\right)$, then:*

$$\min_{t=1,\ldots,T}\|\nabla f(\mathbf{w}_t)\|^2 \leq \mathcal{O}\left(\frac{\rho_{n,d}\left(f(\mathbf{w}_1) - f^\star\right)}{(mn)^{1/3}}\right) + \mathcal{O}\left(\frac{d\rho_{n,d}^2}{(mn)^{1/3}}\right),$$

*with probability at least $1 - \mathcal{O}\left(1/(\rho_{n,d}n)\right)$.*

When $m = \sqrt{n}$, mini-batch SAGD achieves the convergence of rate $\mathcal{O}(1/\sqrt{n})$. When $m = n$, i.e., in the full batch setting, Theorem 5 recovers SAGD's convergence rate $\mathcal{O}(1/n^{2/3})$. In terms of computational complexity, the mini-batch SAGD requires $\mathcal{O}(m^{4/3}n^{1/3})$ stochastic gradient computations for $\mathcal{O}(m^{4/3}/n^{2/3})$ passes over $n$ samples, while SAGD requires $\mathcal{O}(n^{5/3})$ stochastic gradient computations. Thus, the mini-batch SAGD has the advantage of decreasing the computation complexity, but displays a slower convergence than SAGD.

## 4 Numerical Experiments

In this section, we evaluate our proposed mini-batch SAGD algorithm on various deep learning models against popular optimization methods: SGD with momentum [29], Adam [19], RMSprop [35], and Adabound [24]. We consider three tasks: the classification tasks on MNIST [22] and CIFAR-10 [20], and the language modeling task on Penn Treebank [25] and the SNLI dataset [3], corpus of $570\,000$ human-written English sentence pairs where the goal is to predict if an hypothesis is an *entailment*, *contradiction* or *neutral* with respect to a given text.

The setup of each task is given in the following table:

| Dataset | Network Type | Architectures |
| --- | --- | --- |
| MNIST | Feedforward | 2-Layer with ReLU and 2-Layer with Sigmoid |
| CIFAR-10 | Deep Convolutional | VGG-19 and ResNet-18 |
| Penn Treebank | Recurrent | 2-Layer LSTM and 3-Layer LSTM |
| SNLI | Recurrent | bidirectional LSTM |

### 4.1 Environmental Settings

**Datasets and Evaluation Metrics:** The MNIST dataset has a training set of 60000 examples and a test set of 10000 examples. The CIFAR-10 dataset consists of 50000 training images and 10000 test images. The Penn Treebank dataset contains 929589, 73760, and 82430 tokens for training, validation, and test, respectively. To better understand the generalization ability of each optimization algorithm with an increasing training sample size $n$, for each task, we construct multiple training sets of different size by sampling from the original training set. For MNIST, training sets of size $n \in \{50, 100, 200, 500, 10^3, 2.10^3, 5.10^3, 10^4, 2.10^4, 5.10^4\}$ are constructed. For CIFAR10, training sets of size $n \in \{200, 500, 10^3, 2.10^3, 5.10^3, 10^4, 2.10^4, 3.10^4, 5.10^4\}$ are constructed. For each $n$, we train the model and report the loss and accuracy on the test set. For Penn Treebank and SNLI, all training samples are used to train the model and we report the training perplexity and the test perplexity across epochs. Cross-entropy is used as the loss function throughout experiments. The mini-batch size is set to be 128 for CIFAR10 and MNIST, 20 for Penn Treebank and SNLI. We repeat each experiment 5 times and report the mean of the results.

**Hyper-parameter setting:** Optimization hyper-parameters affect the quality of solutions. Particularly, [39] highlight that the initial stepsize and the scheme of decaying stepsizes have a considerable impact on the performance. We use grid search method to tune the step size for each optimizer. We

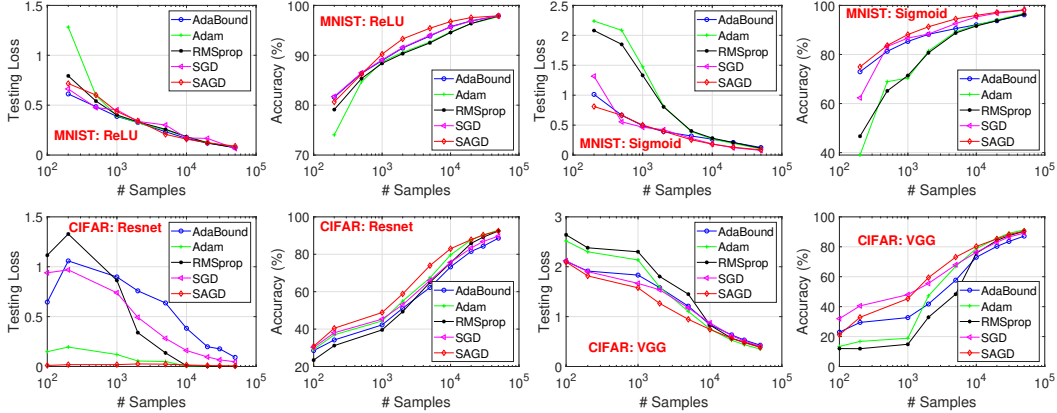

Figure 1: Test loss and test accuracy of ReLU neural network and Sigmoid neural network on MNIST. In both cases, SAGD obtains the best test accuracy among all the methods. Test loss and accuracy of ResNet-18 and VGG-19 on CIFAR10. SAGD achieves the lowest test loss. For VGG-19, SAGD achieves the best test accuracy among all the methods. The x-axis is the number of train samples, and the y-axis is the testing loss/accuracy.

specify the strategy of decaying step sizes and the noise parameter $\sigma$ in the subsections of each task. Parameters $\nu$, $\beta_2$, and $T$ follow the default settings as adaptive algorithms such as RMSprop.

## 4.2   Numerical results

**Feedforward Neural Network.** For image classification on MNIST, we focus on two 2-layer fully connected neural networks with either ReLU or Sigmoid activation functions. We run 100 epochs and decay the learning rate by 0.5 every 30 epochs. We use $\sigma = 0.8$ for ReLU and $\sigma = 1.0$ for Sigmoid. Figure 1 presents the loss and accuracy on the test set given different training set sizes. Since all algorithms attain the 100% training accuracy, the performance on the training set is omitted. Figure 1 shows that, for ReLU neural network, SAGD performs slightly better than the other algorithms in terms of test accuracy. Figure 1 also presents the results on Sigmoid neural network where SAGD achieves the best test accuracy among all the algorithms.

**Convolutional Neural Network.** We use ResNet-18 [17] and VGG-19 [34] for the CIFAR-10 image classification task. We run 200 epochs and decay the learning rate by 0.1 every 30 epochs. We use $\sigma = 0.01$ for both ResNet-18 and VGG-19. Figure 1 shows that SAGD has higher test accuracy than the other algorithms. Figure 1 also reports the results on VGG-19. SAGD performs similarly to SGD and achieves a higher test loss than the other adaptive gradient methods. Adam and our method performs better than the other adaptive gradient algorithms when sample size is large regarding the test accuracy.

**Recurrent Neural Network.** An experiment on Penn Treebank is conducted for the language modeling task with 2-layer Long Short-Term Memory (LSTM) [26] network and 3-layer LSTM. We use $\sigma = 0.01$ for both models. We train them for a fixed budget of 500 epochs and omit the learning-rate decay. Perplexity is used as the metric to evaluate the performance and learning curves are plotted in Figure 2. For a 2-layer LSTM, we observe in Figure 2 that RMSprop and Adam achieve a lower training perplexity than SAGD. However, SAGD performs the best in terms of the test perplexity. In particular, we observe that after 200 epochs, the test perplexity of RMSprop and Adam starts increasing, but the training perplexity continues decreasing (over-fitting occurs). For the 3-layer LSTM, Figure 2 shows that the perplexity of Adam and RMSprop start increasing significantly after 150 epochs (*over-fitting*) while the perplexity of SAGD keeps decreasing. SAGD, SGD and AdaBound perform better than Adam, and RMSprop in terms of over-fitting. Table 1 shows the best test perplexity of 2-layer LSTM and 3-layer LSTM for all the algorithms.

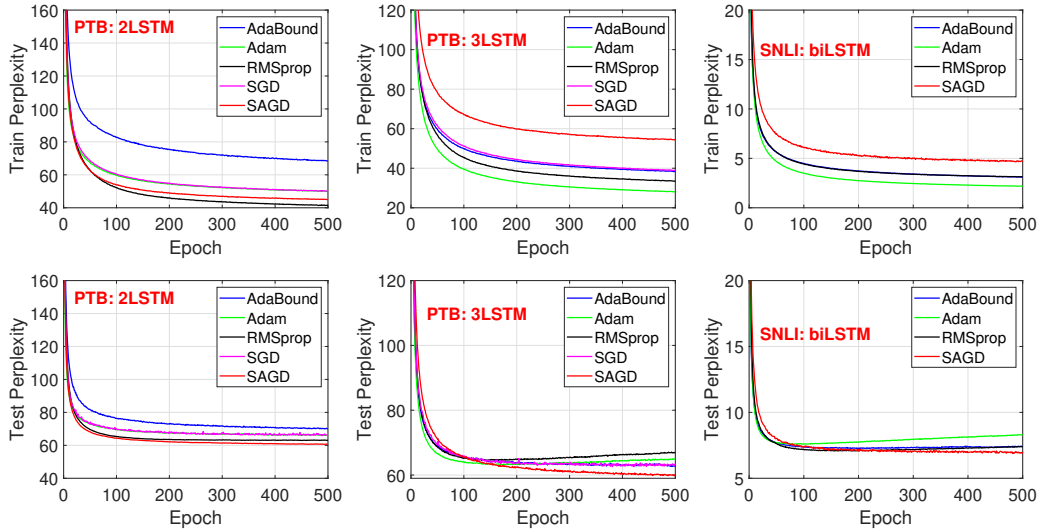

Figure 2: Train (upper panels) and test (bottom panels) perplexity of 2-layer LSTM (2LSTM), 3-layer LSTM (3LSTM) and biLSTM. Even though some baseline optimizers achieve better training performance than SAGD, the latter performs the best in terms of test perplexity among all methods.

Table 1: Test Perplexity of LSTMs on Penn Treebank. Bold number indicates the best result.

|  | RMSprop | Adam | AdaBound | SGD | SAGD |
|---|---|---|---|---|---|
| 2-layer LSTM | $62.87 \pm 0.05$ | $66.02 \pm 0.05$ | $65.82 \pm 0.08$ | $65.96 \pm 0.23$ | **$60.66 \pm 0.05$** |
| 3-layer LSTM | $63.97 \pm 018$ | $63.23 \pm 004$ | $62.33 \pm 0.07$ | $62.51 \pm 0.11$ | **$59.43 \pm 0.24$** |

**Bidirectional LSTM.** We use a bi-directional LSTM architecture, as the concatenation of a forward LSTM and a backward LSTM as described in [9]. We use 300 dimensions as fixed word embeddings and set the learning rate following the method described above. We set noise parameter $\sigma = 0.01$. In Figure 2, we compare mini-batch SAGD to the following baselines: Adam [19], RMSprop [35], and Adabound [24]. As in the language modeling task on PTB, we observe that whilst SAGD displays a worse loss perplexity than baseline methods, it keeps a low testing perplexity through the epochs. This phenomena has been observed in all of our experiments and highlights the advantage of our proposed method to present *reused* samples to the model as if they were fresh ones. Thus, over-fitting is less likely to happen and testing loss will remain low. For instance, Adam achieves the best training perplexity, yet displays an increasing testing perplexity after a few epochs, which leads to low test accuracy.

## 5 Conclusion

In this paper, we focus on the generalization ability of adaptive gradient methods. Concerned with the observation that adaptive gradient methods generalize worse than SGD for over-parameterized neural networks and given the limited theoretical understanding of the generalization of those methods, we propose **S**table **A**daptive **G**radient **D**escent (SAGD) methods, which boost the generalization performance in both theory and practice through a novel use of differential privacy. The proposed algorithms generalize well with provable high-probability convergence bounds of the population gradient. Experimental studies highlight that the proposed algorithms are competitive and often better than baseline algorithms for training deep neural networks and demonstrate the aptitude of our method to avoid over-fitting through a differential privacy mechanism.

## Broader Impact

We believe that our work stands in the line of several papers towards improving generalization and avoiding over-fitting. Indeed, the basic principle of our method is to fit any given model, in particular deep model, using an intermediate differentially-private mechanisms allowing the model to fit fresh samples while passing over the same batch of $n$ observations. The impact of such work is straightforward and could avoid learning, and thus reproducing at testing phase, the bias existent in the training dataset.

## Acknowledgments and Disclosure of Funding

We thank the anonymous Referees and Area Chair for their constructive comments. The work is supported by Baidu Research.

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
