[Supplementary Material]

# A  Differential Privacy and Generalization Analysis

## A.1  Proof of Lemma 1

By applying Theorem 8 from [11] to gradient computation, we obtain Lemma 1.

**Lemma 1.** *Let $\mathcal{A}$ be an $(\epsilon, \delta)$-differentially private gradient descent algorithm with access to training set $S$ of size $n$. Let $\mathbf{w}_t = \mathcal{A}(S)$ be the parameter generated at iteration $t \in [T]$ and $\hat{\mathbf{g}}_t$ the empirical gradient on $S$. For any $\sigma > 0$, $\beta > 0$, if the privacy cost of $\mathcal{A}$ satisfies $\epsilon \leq \sigma/13$, $\delta \leq \sigma\beta/(26\ln(26/\sigma))$, and sample size $n \geq 2\ln(8/\delta)/\epsilon^2$, we then have*

$$\mathbb{P}\left\{|\hat{\mathbf{g}}_t^i - \mathbf{g}_t^i| \geq G\sigma\right\} \leq \beta, \quad \forall i \in [d] \text{ and } \forall t \in [T].$$

**Proof** Theorem 8 in [11] shows that in order to achieve generalization error $\tau$ with probability $1 - \rho$ for an $(\epsilon, \delta)$-differentially private algorithm (i.e., in order to guarantee for every function $\phi_t$, $\forall t \in [T]$, we have $\mathbb{P}\left[|\mathcal{P}\left[\phi_t\right] - \mathcal{E}_S\left[\phi_t\right]| \geq \tau\right] \leq \rho$), where $\mathcal{P}\left[\phi_t\right]$ is the population value, $\mathcal{E}_S\left[\phi_t\right]$ is the empirical value evaluated on $S$ and $\rho$ and $\tau$ are any positive constant, we can set the $\epsilon \leq \frac{\tau}{13}$ and $\delta \leq \frac{\tau\rho}{26\ln(26/\tau)}$. In our context, $\tau = \sigma$, $\beta = \rho$, $\phi_t$ is the gradient computation function $\nabla\ell(\mathbf{w}_t, \mathbf{z})$, $\mathcal{P}\left[\phi_t\right]$ represents the population gradient $\mathbf{g}_t^i/G$, $\forall i \in [p]$, and $\mathcal{E}_S\left[\phi_t\right]$ represents the sample gradient $\hat{\mathbf{g}}_t^i/G$, $\forall i \in [p]$. Thus we have $\mathbb{P}\left\{\left|\hat{\mathbf{g}}_t^i - \mathbf{g}_t^i\right|/G \geq \tau\right\} \leq \rho$ if $\epsilon \leq \frac{\sigma}{13}, \delta \leq \frac{\sigma\beta}{26\ln(26/\sigma)}$.

## A.2  Proof of Lemma 2

**Lemma 2.** SAGD *with* DPG-LAP *(Alg. 1) is $(\frac{\sqrt{T\ln(1/\delta)}G_1}{n\sigma}, \delta)$-differentially private.*

**Proof** At each iteration $t$, the algorithm is composed of two sequential parts: DPG to access the training set $S$ and compute $\tilde{\mathbf{g}}_t$, and parameter update based on estimated $\tilde{\mathbf{g}}_t$. We mark the DPG as part $\mathcal{A}$ and the gradient descent as part $\mathcal{B}$. We first show $\mathcal{A}$ preserves $\frac{G_1}{n\sigma}$-differential privacy. Then according to the *post-processing property* of differential privacy (Proposition 2.1 in [14]) we have $\mathcal{B} \circ \mathcal{A}$ is also $\frac{G_1}{n\sigma}$-differentially private.

The part $\mathcal{A}$ (DPG-Lap) uses the basic tool from differential privacy, the "Laplace Mechanism" (Definition 3.3 in [14]). The Laplace Mechanism adds i.i.d. Laplace noise to each coordinate of the output. Adding noise from $Lap(\sigma)$ to a query of $G_1/n$ sensitivity preserves $G_1/n\sigma$-differential privacy by (Theorem 3.6 in [14]). Over $T$ iterations, we have $T$ applications of a DPG-Lap. By the advanced composition theorem (Theorem 3.20 in [14]), $T$ applications of a $\frac{G_1}{n\sigma}$-differentially private algorithm is $(\frac{\sqrt{T\ln(1/\delta)}G_1}{n\sigma}, \delta)$-differentially private. So SAGD with DPG-Lap is $(\frac{\sqrt{T\ln(1/\delta)}2G_1}{n\sigma}, \delta)$-differentially private. $\qquad\square$

## A.3  Proof of Theorem 1

> **Theorem 1** *Given $\sigma > 0$, let $\tilde{\mathbf{g}}_1, ..., \tilde{\mathbf{g}}_T$ be gradients computed by DPG-LAP in SAGD. Set the number of iterations $2n\sigma^2/G_1^2 \leq T \leq n^2\sigma^4/(169\ln(1/(\sigma\beta))G_1^2)$, then for $t \in [T]$, $\beta > 0$, $\mu > 0$:*
> $$\mathbb{P}\left\{\|\tilde{\mathbf{g}}_t - \mathbf{g}_t\| \geq \sqrt{d}\sigma(G + \mu)\right\} \leq d\beta + d\exp(-\mu).$$

**Proof** The concentration bound is decomposed into two parts:

$$\mathbb{P}\left\{\|\tilde{\mathbf{g}}_t - \mathbf{g}_t\| \geq \sqrt{d}\sigma(G + \mu)\right\} \leq \underbrace{\mathbb{P}\left\{\|\tilde{\mathbf{g}}_t - \hat{\mathbf{g}}_t\| \geq \sqrt{d}\sigma\mu\right\}}_{T_1:\text{ empirical error}} + \underbrace{\mathbb{P}\left\{\|\hat{\mathbf{g}}_t - \mathbf{g}_t\| \geq \sqrt{d}\sigma\right\}}_{T_2:\text{ generalization error}}.$$

In the above inequality, there are two types of errors which we need to control. The first type of error, referred to as empirical error $T_1$, is the deviation between the differentially private estimated gradient $\tilde{\mathbf{g}}_t$ and the empirical gradient $\hat{\mathbf{g}}_t$. The second type of error, referred to as generalization error $T_2$, is the deviation between the empirical gradient $\hat{\mathbf{g}}_t$ and the population gradient $\mathbf{g}_t$.

The second term $T_2$ can be bounded thorough the generalization guarantee of differential privacy. Recall that from Lemma 1, under the condition in Theorem 3, we have for all $t \in [T]$, $i \in [d]$:

$$\mathbb{P}\left\{|\hat{\mathbf{g}}_t^i - \mathbf{g}_t^i| \geq G\sigma\right\} \leq \beta.$$

So that we have

$$\mathbb{P}\left\{\|\hat{\mathbf{g}}_t - \mathbf{g}_t\| \geq \sqrt{d}G\sigma\right\} \leq \mathbb{P}\left\{\|\hat{\mathbf{g}}_t - \mathbf{g}_t\|_\infty \geq G\sigma\right\} \leq d\mathbb{P}\left\{|\hat{\mathbf{g}}_t^i - \mathbf{g}_t^i| \geq G\sigma\right\} \leq d\beta. \quad (3)$$

Now we bound the second term $T_1$. Recall that $\tilde{\mathbf{g}}_t = \hat{\mathbf{g}}_t + \mathbf{b}_t$, where $\mathbf{b}_t$ is a noise vector with each coordinate drawn from Laplace noise $\text{Lap}(\sigma)$. In this case, we have

$$\mathbb{P}\left\{\|\tilde{\mathbf{g}}_t - \hat{\mathbf{g}}_t\| \geq \sqrt{d}\sigma\mu\right\} \leq \mathbb{P}\left\{\|\mathbf{b}_t\| \geq \sqrt{d}\sigma\mu\right\} \leq \mathbb{P}\left\{\|\mathbf{b}_t\|_\infty \geq \sigma\mu\right\} \leq d\mathbb{P}\left\{|\mathbf{b}_t^i| \geq \sigma\mu\right\} = d\exp(-\mu).$$

$$(4)$$

The second inequality comes from $\|\mathbf{b}_t\| \leq \sqrt{d}\|\mathbf{b}_t\|_\infty$. The last equality comes from the property of Laplace distribution. Combine (3) and (4), we complete the proof. □

## A.4 Proof of Lemma 3

**Lemma 3.** SAGD *with* DPG-SPARSE *(Alg. 2) is* $\left(\frac{\sqrt{C_s \ln(2/\delta)}2G_1}{n\sigma}, \delta\right)$*-differentially private.*

**Proof** At each iteration $t$, the algorithm is composed of two sequential parts: DPG-Sparse (part $\mathcal{A}$) and parameter update based on estimated $\tilde{\mathbf{g}}_t$ (part $\mathcal{B}$). We first show $\mathcal{A}$ preserves $\frac{2G_1}{n\sigma}$-differential privacy. Then according to the *post-processing property* of differential privacy (Proposition 2.1 in [14]) we have $\mathcal{B} \circ \mathcal{A}$ is also $\frac{2G_1}{n\sigma}$-differentially private.

The part $\mathcal{A}$ (DPG-Sparse) is a composition of basic tools from differential privacy, the "Sparse Vector Algorithm" (Algorithm 2 in [14]) and the "Laplace Mechanism" (Definition 3.3 in [14]). In our setting, the sparse vector algorithm takes as input a sequence of $T$ sensitivity $G_1/n$ queries, and for each query, attempts to determine whether the value of the query, evaluated on the private dataset $S_1$, is above a fixed threshold $\gamma + \tau$ or below it. In our instantiation, the $S_1$ is the private data set, and each function corresponds to the gradient computation function $\hat{\mathbf{g}}_t$ which is of sensitivity $G_1/n$. By the privacy guarantee of the sparse vector algorithm, the sparse vector portion of SAGD satisfies $G_1/n\sigma$-differential privacy. The Laplace mechanism portion of SAGD satisfies $G_1/n\sigma$-differential privacy by ( Theorem 3.6 in [14]). Finally, the composition of two mechanisms satisfies $\frac{2G_1}{n\sigma}$-differential privacy. For the sparse vector technique, only the query that fails the validation, corresponding to the 'above threshold', release the privacy of private dataset $S_1$ and pays a $\frac{2G_1}{n\sigma}$ privacy cost. Over all the iterations $T$, We have $C_s$ queries fail the validation. Thus, by the advanced composition theorem (Theorem 3.20 in [14]), $C_s$ applications of a $\frac{2G}{n\sigma}$-differentially private algorithm is $\left(\frac{\sqrt{C_s \ln(2/\delta)}2G_1}{n\sigma}, \delta\right)$-differentially private. Therefore, SAGD with DPG-Sparse is $\left(\frac{\sqrt{C_s \ln(2/\delta)}2G_1}{n\sigma}, \delta\right)$-differentially private. □

## A.5 Proof of Theorem 3:

**Theorem 3** *Given $\sigma > 0$, let $\tilde{\mathbf{g}}_1, ..., \tilde{\mathbf{g}}_T$ be the gradients computed by* DPG-SPARSE *in* SAGD. *With a budget $n\sigma^2/(2G_1^2) \leq C_s \leq n^2\sigma^4/(676 \ln(1/(\sigma\beta))G_1^2)$, then for $t \in [T]$, $\beta > 0$, $\mu > 0$:*

$$\mathbb{P}\left\{\|\tilde{\mathbf{g}}_t - \mathbf{g}_t\| \geq \sqrt{d}\sigma(1+\mu)\right\} \leq d\beta + d\exp(-\mu).$$

**Proof** The concentration bound can be decomposed into two parts:

$$\mathbb{P}\left\{\|\tilde{\mathbf{g}}_t - \mathbf{g}_t\| \geq \sqrt{d}\sigma(1+\mu)\right\} \leq \underbrace{\mathbb{P}\left\{\|\tilde{\mathbf{g}}_t - \hat{\mathbf{g}}_{s_1,t}\| \geq \sqrt{d}\sigma\mu\right\}}_{T_1: \text{ empirical error}} + \underbrace{\mathbb{P}\left\{\|\hat{\mathbf{g}}_{s_1,t} - \mathbf{g}_t\| \geq \sqrt{d}\sigma\right\}}_{T_2: \text{ generalization error}},$$

which yields

$$\mathbb{P}\left\{\|\hat{\mathbf{g}}_{s_1,t} - \mathbf{g}_t\| \geq \sqrt{d}\sigma\right\} \leq \mathbb{P}\left\{\|\hat{\mathbf{g}}_{s_1,t} - \mathbf{g}_t\|_\infty \geq \sigma\right\} \leq d\mathbb{P}\left\{|\hat{\mathbf{g}}_{s_1,t}^i - \mathbf{g}_t^i| \geq \sigma\right\} \leq d\beta. \quad (5)$$

Now we bound the second term $T_1$ by considering two cases, by depending on whether DPG-3 answers the query $\tilde{\mathbf{g}}_t$ by returning $\tilde{\mathbf{g}}_t = \hat{\mathbf{g}}_{s_1,t} + \mathbf{v}_t$ or by returning $\tilde{\mathbf{g}}_t = \hat{\mathbf{g}}_{s_2,t}$. In the first case, we have

$$\|\tilde{\mathbf{g}}_t - \hat{\mathbf{g}}_{s_1,t}\| = \|\mathbf{v}_t\|$$

and

$$\mathbb{P}\left\{\|\tilde{\mathbf{g}}_t - \hat{\mathbf{g}}_{s_1,t}\| \geq \sqrt{d}\sigma\mu\right\} = \mathbb{P}\left\{\|\mathbf{v}_t\| \geq \sqrt{d}\sigma\mu\right\} \leq d\exp(-\mu).$$

The last inequality comes from the $\|\mathbf{v}_t\| \leq \sqrt{d}\|\mathbf{v}_t\|_\infty$ and properties of the Laplace distribution.

In the second case, we have

$$\|\tilde{\mathbf{g}}_t - \hat{\mathbf{g}}_{s_1,t}\| = \|\hat{\mathbf{g}}_{s_2,t} - \hat{\mathbf{g}}_{s_1,t}\| \leq |\gamma| + |\tau|$$

and

$$\begin{aligned}
\mathbb{P}\left\{\|\tilde{\mathbf{g}}_t - \hat{\mathbf{g}}_{s_1,t}\| \geq \sqrt{d}\sigma\mu\right\} &= \mathbb{P}\left\{|\gamma| + |\tau| \geq \sqrt{d}\sigma\mu\right\} \\
&\leq \mathbb{P}\left\{|\gamma| \geq \frac{2}{6}\sqrt{d}\sigma\mu\right\} + \mathbb{P}\left\{|\tau| \geq \frac{4}{6}\sqrt{d}\sigma\mu\right\} \\
&= 2\exp(-\sqrt{d}\mu/6).
\end{aligned}$$

Combining these two cases, we have

$$\begin{aligned}
\mathbb{P}\left\{\|\tilde{\mathbf{g}}_t - \hat{\mathbf{g}}_{s_1,t}\| \geq \sqrt{d}\sigma\mu\right\} &\leq \max\left\{\mathbb{P}\left\{\|\mathbf{v}_t\| \geq \sqrt{d}\sigma\mu\right\}, \mathbb{P}\left\{|\gamma| + |\tau| \geq \sqrt{d}\sigma\mu\right\}\right\} \\
&\leq \max\left\{d\exp(-\mu), 2\exp(-\sqrt{d}\mu/6)\right\} \\
&= d\exp(-\mu). \quad (6)
\end{aligned}$$

We complete the proof by combining (5) and (6).

$\square$

## B   Non-asymptotic Convergence analysis

In this section, we present the proofs for Theorems 2, 4 , 5.

### B.1   Proof of Theorem 2 and Theorem 4

The proof of Theorem 2 consists of two parts: We first prove that the convergence rate of a gradient-based iterative algorithm is related to the gradient concentration error $\alpha$ and its iteration time $T$. Then we combine the concentration error $\alpha$ achieved by SAGD with DPG-Lap in Theorem 1 with the first part to complete the proof of Theorem 2. To simplify the analysis, we first use $\alpha$ and $\xi$ to denote the generalization error $\sqrt{d}\sigma(G + \mu)$ and probability $d\beta + d\exp(-\mu)$ in Theorem 1 in the following analysis. The details are presented in the following theorem.

---

**Theorem 6** *Let $\tilde{\mathbf{g}}_1, ..., \tilde{\mathbf{g}}_T$ be the noisy gradients generated in Algorithm 1 through DPG oracle over $T$ iterations. Then, for every $t \in [T]$, $\tilde{\mathbf{g}}_t$ satisfies*

$$\mathbb{P}\{\|\tilde{\mathbf{g}}_t - \mathbf{g}_t\| \geq \alpha\} \leq \xi,$$

*where the values of $\alpha$ and $\xi$ are given in Section A.*

---

With the guarantee of Theorem 6, we have the next theorem which shows the convergence of SAGD.

**Theorem 7** *Let $\eta_t = \eta$. Further more assume that $\nu$, $\beta$ and $\eta$ are chosen such that the following conditions satisfied: $\eta \leq \frac{\nu}{2L}$. Under the Assumption A1 and A2, the Algorithm 1 with $T$ iterations, $\phi_t(\tilde{\mathbf{g}}_1, ..., \tilde{\mathbf{g}}_t) = \tilde{\mathbf{g}}_t$ and $\mathbf{v}_t = (1 - \beta_2) \sum_{i=1}^t \beta_2^{t-i} \tilde{\mathbf{g}}_i^2$ achieves:*

$$\min_{t=1,...,T} \|\nabla f(x_t)\|^2 \leq (G + \nu) \times \left( \frac{f(\mathbf{w}_1) - f^\star}{\eta T} + \frac{3\alpha^2}{4\nu} \right), \tag{7}$$

*with probability at least $1 - T\xi$.*

We can now tackle the proof of our result stated in Theorem 7.

**Proof** Using the update rule of RMSprop, we have $\phi_t(\tilde{\mathbf{g}}_1, ..., \tilde{\mathbf{g}}_t) = \tilde{\mathbf{g}}_t$ and $\psi_t(\tilde{\mathbf{g}}_1, ..., \tilde{\mathbf{g}}_t) = (1 - \beta_2) \sum_{i=1}^t \beta_2^{t-i} \tilde{\mathbf{g}}_i^2$. Thus, we can rewrite the update of Algorithm 1 as:

$$\mathbf{w}_{t+1} = \mathbf{w}_t - \eta_t \tilde{\mathbf{g}}_t / (\sqrt{\mathbf{v}_t} + \nu) \text{ and } \mathbf{v}_t = (1 - \beta_2) \sum_{i=1}^t \beta_2^{t-i} \tilde{\mathbf{g}}_i^2.$$

Let $\Delta_t = \tilde{\mathbf{g}}_t - g_t$, we obtain:

$$f(\mathbf{w}_{t+1})$$

$$\leq f(\mathbf{w}_t) + \langle \mathbf{g}_t, \mathbf{w}_{t+1} - \mathbf{w}_t \rangle + \frac{L}{2} \|\mathbf{w}_{t+1} - \mathbf{w}_t\|^2$$

$$= f(\mathbf{w}_t) - \eta_t \langle \mathbf{g}_t, \tilde{\mathbf{g}}_t / (\sqrt{\mathbf{v}_t} + \nu) \rangle + \frac{L\eta_t^2}{2} \left\| \frac{\tilde{\mathbf{g}}_t}{(\sqrt{\mathbf{v}_t} + \nu)} \right\|^2$$

$$= f(\mathbf{w}_t) - \eta_t \left\langle \mathbf{g}_t, \frac{\mathbf{g}_t + \Delta_t}{\sqrt{\mathbf{v}_t} + \nu} \right\rangle + \frac{L\eta_t^2}{2} \left\| \frac{\mathbf{g}_t + \Delta_t}{\sqrt{\mathbf{v}_t} + \nu} \right\|^2$$

$$\leq f(\mathbf{w}_t) - \eta_t \left\langle \mathbf{g}_t, \frac{\mathbf{g}_t}{\sqrt{\mathbf{v}_t} + \nu} \right\rangle - \eta_t \left\langle \mathbf{g}_t, \frac{\Delta_t}{\sqrt{\mathbf{v}_t} + \nu} \right\rangle + L\eta_t^2 \left( \left\| \frac{\mathbf{g}_t}{\sqrt{\mathbf{v}_t} + \nu} \right\|^2 + \left\| \frac{\Delta_t}{\sqrt{\mathbf{v}_t} + \nu} \right\|^2 \right)$$

$$= f(\mathbf{w}_t) - \eta_t \sum_{i=1}^d \frac{[\mathbf{g}_t]_i^2}{\sqrt{\mathbf{v}_t^i} + \nu} - \eta_t \sum_{i=1}^d \frac{\mathbf{g}_t^i \Delta_t^i}{\sqrt{\mathbf{v}_t^i} + \nu} + L\eta_t^2 \left( \sum_{i=1}^d \frac{[\mathbf{g}_t]_i^2}{(\sqrt{\mathbf{v}_t^i} + \nu)^2} + \sum_{i=1}^d \frac{[\Delta_t]_i^2}{(\sqrt{\mathbf{v}_t^i} + \nu)^2} \right)$$

$$\leq f(\mathbf{w}_t) - \eta_t \sum_{i=1}^d \frac{[\mathbf{g}_t]_i^2}{\sqrt{\mathbf{v}_t^i} + \nu} + \frac{\eta_t}{2} \sum_{i=1}^d \frac{[\mathbf{g}_t]_i^2 + [\Delta_t]_i^2}{\sqrt{\mathbf{v}_t^i} + +\nu} + \frac{L\eta_t^2}{\nu} \left( \sum_{i=1}^d \frac{[\mathbf{g}_t]_i^2}{\sqrt{\mathbf{v}_t^i} + \nu} + \sum_{i=1}^d \frac{[\Delta_t]_i^2}{\sqrt{\mathbf{v}_t^i} + \nu} \right)$$

$$= f(\mathbf{w}_t) - \left( \eta_t - \frac{\eta_t}{2} - \frac{L\eta_t^2}{\nu} \right) \sum_{i=1}^d \frac{[\mathbf{g}_t]_i^2}{\sqrt{\mathbf{v}_t^i} + \nu} + \left( \frac{\eta_t}{2} + \frac{L\eta_t^2}{\nu} \right) \sum_{i=1}^d \frac{[\Delta_t]_i^2}{\sqrt{\mathbf{v}_t^i} + \nu}.$$

Given the parameter setting from the theorem, we see the following condition hold:

$$\frac{L\eta_t}{\nu} \leq \frac{1}{4}.$$

Then we obtain

$$f(\mathbf{w}_{t+1}) \leq f(\mathbf{w}_t) - \frac{\eta}{4} \sum_{i=1}^d \frac{[\mathbf{g}_t]_i^2}{\sqrt{\mathbf{v}_t^i} + \nu} + \frac{3\eta}{4} \sum_{i=1}^d \frac{[\Delta_t]_i^2}{\sqrt{\mathbf{v}_t^i} + \nu}$$

$$\leq f(\mathbf{w}_t) - \frac{\eta}{G + \nu} \|\mathbf{g}_t\|^2 + \frac{3\eta}{4\epsilon} \|\Delta_t\|^2.$$

The second inequality follows from the fact that $0 \leq \mathbf{v}_t^i \leq G^2$. Using the telescoping sum and rearranging the inequality, we obtain

$$\frac{\eta}{G + \nu} \sum_{t=1}^T \|\mathbf{g}_t\|^2 \leq f(\mathbf{w}_1) - f^\star + \frac{3\eta}{4\epsilon} \sum_{t=1}^T \|\Delta_t\|^2.$$

Multiplying with $\frac{G+\nu}{\eta T}$ on both sides and with the guarantee in Theorem 1 that $\|\Delta_t\| \leq \alpha$ with probability at least $1 - \xi$, we obtain

$$\min_{t=1,\ldots,T} \|\mathbf{g}_t\|^2 \leq (G + \nu) \times \left( \frac{f(\mathbf{w}_1) - f^\star}{\eta T} + \frac{3\alpha^2}{4\nu} \right) ,$$

with probability at least $1 - T\xi$. $\qquad\square$

We now present the proof of our Theorem 2.

---

**Theorem 2** *Given training set $S$ of size $n$, for $\nu > 0$, if $\eta_t = \eta$ with $\eta \leq \nu/(2L)$, $\sigma = 1/n^{1/3}$, iteration number $T = n^{2/3} / \left( 169 G_1^2 (\ln d + 7 \ln n/3) \right)$, $\mu = \ln(1/\beta)$ and $\beta = 1/(dn^{5/3})$, then SAGD with DPG-LAP algorithm yields:*

$$\min_{1 \leq t \leq T} \|\nabla f(\mathbf{w}_t)\|^2 \leq \mathcal{O}\left( \frac{\rho_{n,d}\, (f(\mathbf{w}_1) - f^\star)}{n^{2/3}} \right) + \mathcal{O}\left( \frac{d\rho_{n,d}^2}{n^{2/3}} \right) ,$$

*with probability at least $1 - \mathcal{O}\left( 1/(\rho_{n,d} n) \right)$.*

---

**Proof** First consider the gradient concentration bound achieved by SAGD (Theorem 1 and Theorem 3) that if $\frac{2n\sigma^2}{G_1^2} \leq T \leq \frac{n^2\sigma^4}{169 \ln(1/(\sigma\beta)) G_1^2}$, we have

$$\mathbb{P}\left\{ \|\tilde{\mathbf{g}}_t - \mathbf{g}_t\| \geq \sqrt{d}\sigma(G + \mu) \right\} \leq d\beta + d\exp(-\mu), \;\; \forall t \in [T].$$

Then bring the setting in Theorem 2 that $\sigma = 1/n^{1/3}$, let $\mu = \ln(1/\beta)$ and $\beta = 1/(dn^{5/3})$, we have

$$\|\tilde{\mathbf{g}}_t - \mathbf{g}_t\|^2 \leq d(1 + \ln d + \frac{5}{3} \ln n)^2 / n^{2/3} ,$$

with probability at least $1 - 1/n^{5/3}$, when we set $T = n^{2/3} / \left( 169 G_1^2 (\ln d + \frac{7}{3} \ln n) \right)$.

Connect this result with Theorem 7, so that we have $\alpha^2 = d(1 + \ln d + \frac{5}{3} \ln n)^2 / n^{2/3}$ and $\xi = 1/n^{5/3}$. Bring the value $\alpha^2$, $\xi$ and $T = n^{2/3} / \left( 169 G_1^2 (\ln d + \frac{7}{3} \ln n) \right)$ into (7), with $\rho_{n,d} = O\left( \ln n + \ln d \right)$, we have

$$\min_{t=1,\ldots,T} \|\nabla f(\mathbf{w}_t)\|^2 \leq O\left( \frac{\rho_{n,d}\, (f(\mathbf{w}_1) - f^\star)}{n^{2/3}} \right) + O\left( \frac{d\rho_{n,d}^2}{n^{2/3}} \right) ,$$

with probability at least $1 - O\left( \frac{1}{\rho_{n,d} n} \right)$ which concludes the proof. $\qquad\square$

---

**Theorem 4** *Given training set $S$ of size $n$, for $\nu > 0$, if $\eta_t = \eta$ which are chosen with $\eta \leq \nu/(2L)$, noise level $\sigma = 1/n^{1/3}$, and iteration number $T = n^{2/3} / \left( 676 G_1^2 (\ln d + \frac{7}{3} \ln n) \right)$, then SAGD with DPG-SPARSE algorithm yields:*

$$\min_{1 \leq t \leq T} \|\nabla f(\mathbf{w}_t)\|^2 \leq \mathcal{O}\left( \frac{\rho_{n,d}\, (f(\mathbf{w}_1) - f^\star)}{n^{2/3}} \right) + \mathcal{O}\left( \frac{d\rho_{n,d}^2}{n^{2/3}} \right) ,$$

*with probability at least $1 - \mathcal{O}\left( 1/(\rho_{n,d} n) \right)$.*

---

**Proof** The proof of Theorem 4 follows the proof of Theorem 2 by considering the case $C_s = T$. $\quad\square$

## B.2 Proof of Theorem 5

> **Theorem 5** *Consider the mini-batch* SAGD *with* DPG-LAP. *Given $S$ of size $n$, with $\nu > 0$, $\eta_t = \eta \leq \nu/(2L)$, noise level $\sigma = 1/(mn)^{1/6}$, and epoch $T = m^{4/3}/\left(n^{2/3}169G_1^2(\ln d + \frac{7}{3}\ln n)\right)$, then:*
>
> $$\min_{t=1,\ldots,T}\|\nabla f(\mathbf{w}_t)\|^2 \leq \mathcal{O}\left(\frac{\rho_{n,d}\left(f(\mathbf{w}_1)-f^\star\right)}{(mn)^{1/3}}\right) + \mathcal{O}\left(\frac{d\rho_{n,d}^2}{(mn)^{1/3}}\right),$$
>
> *with probability at least $1 - \mathcal{O}\left(1/(\rho_{n,d}n)\right)$.*

**Proof** When mini-batch SAGD calls **DPG** to access each batch $s_k$ with size $m$ for $T$ times, we have mini-batch SAGD preserves $(\frac{\sqrt{T\ln(1/\delta)}G_1}{m\sigma}, \delta)$-deferential privacy for each batch $s_k$. Now consider the gradient concentration bound achieved by DPG-Lap (Theorem 1) that if $\frac{2m\sigma^2}{G_1^2} \leq T \leq \frac{m^2\sigma^4}{169\ln(1/(\sigma\beta))G_1^2}$, we have

$$\mathbb{P}\left\{\|\tilde{\mathbf{g}}_t - \mathbf{g}_t\| \geq \sqrt{d}\sigma(G+\mu)\right\} \leq d\beta + d\exp(-\mu), \ \forall t \in [T].$$

Then bring the setting in Theorem 5 that $\sigma = 1/(nm)^{1/6}$, let $\mu = \ln(1/\beta)$ and $\beta = 1/(dn^{5/3})$, we have

$$\|\tilde{\mathbf{g}}_t - \mathbf{g}_t\|^2 \leq d(1 + \ln d + \frac{5}{3}\ln n)^2/n^{2/3},$$

with probability at least $1 - 1/n^{5/3}$, when we set epoch $T = m^{4/3}/\left(n^{2/3}169G_1^2\left(\ln d + \frac{7}{3}\ln n\right)\right)$.

Connect this result with Theorem 7, so that we have $\alpha^2 = d(1 + \ln d + \frac{5}{3}\ln n)^2/(mn)^{1/3}$ and $\xi = 1/n^{5/3}$. Bring the value $\alpha^2$, $\xi$ and total iteration number to be $T \times n/m$ with $T = (mn)^{1/3}/\left(169G_1^2(\ln d + \frac{7}{3}\ln n)\right)$ into (7), with $\rho_{n,d} = O(\ln n + \ln d)$, we have

$$\min_{t=1,\ldots,T}\|\nabla f(\mathbf{w}_t)\|^2 \leq O\left(\frac{\rho_{n,d}\left(f(\mathbf{w}_1)-f^\star\right)}{(mn)^{1/3}}\right) + O\left(\frac{d\rho_{n,d}^2}{(mn)^{1/3}}\right),$$

with probability at least $1 - O\left(\frac{1}{\rho_{n,d}n}\right)$. Here we complete the proof. $\qquad\square$