[Reviews · NeurIPS 2020]

Review 1

Summary and Contributions: The paper studies the use of differential privacy as a way to enforce generalization guarantees in the adaptive gradient based optimization methods. There has been an overwhelming flurry of papers on adaptive gradient based optimization recently. Numerous variants have been proposed, but none stand our clearly and consistently. In particular, there has been concern that these newer adaptive methods don't even consistently outperform stochastic gradient descent when it come to generalization error in machine learning setting (rather than optimization error on the training objective itself). The paper employs two natural differentially private algorithms for perturbing the gradients of the objective functions. Via the known generalization guarantees of differential privacy, this implies that the gradients encountered by the algorithm behave like population gradients even if the optimizer makes multiple passes over the data. The main results of the paper show convergence guarantees to a stationary point of the population objective. Both differentially private methods roughly guarantee that the squared norm of the gradient shrinks as O(1/n^{2/3}). Re rebuttal: The author's response only lightly touched on my review and it did neither increase nor decrease my score.

Strengths: This is a natural use of differential privacy for gradient optimization. In particular the use of Thresholdout to filter large gradients is natural and it's good to have a reference for this approach. The analyses seem sound and the proofs are relatively short and clean. The empirical results look promising.

Weaknesses: In the best case, this is a great idea that will lead to a more principled way to make sure that adaptive gradient methods generalize well. However, a more realistic scenario is that this adds yet another set of heuristics to the flurry of adaptive methods with no clear winner. This to some extent an issue with the research area. Despite thousands of papers on this topic there are relatively few stable facts. This paper does not necessarily reduce the entropy. The techniques involved are fairly standard at this point. I had a really hard time reading the plots in Figure 2. But if I see correctly, SAGD gets the lowest test perplexity in both cases, which is nice. I actually took some issue with the last sentence of the broader impact section. This does not seem to be well thought out or supported by empirical facts.

Correctness: The bounds look reasonable given the techniques that are involved.

Clarity: The clarity could be improved.

Relation to Prior Work: Yes.

Reproducibility: No

Additional Feedback: There is no code for the experimental evaluation.


Review 2

Summary and Contributions: The authors propose a new optimization algorithm called Stable Adaptive Gradient Descent (SAGD) that uses ideas from differential privacy and adaptive data analysis to improve the generalization performance of adaptive gradient algorithms. The authors provide theoretical analysis to show that both SAGD and minibatch SAGD converges to the population stationary point with high probability. Experimental results on neural networks are also provided. ======================================================================== I thank the authors for their response. Unfortunately, one of my main concerns (about the low numbers on their ResNet18 and VGG19 baselines) still remain after the rebuttal. While the authors mention in their rebuttal that proper learning rate tuning was done, this does not explain why the numbers obtained are lower than what is standard for these models. On another look at the experimental section of the paper, I think the low numbers are explained because the authors train their models only for 100 epochs, instead of the standard 200 epochs. This does make me less sure about the experimental results provided in the paper, and whether any improvements shown of the proposed algorithm actually scale to standard training setups. I am therefore reducing my score from 7 to a 6.

Strengths: Overall I really like the main idea and its analysis presented in the paper. Being able to simulate gradients that are close to the population gradient whp throughout the course of optimization seems really interesting. This would particularly be useful for smaller datasets where regularization becomes all the more important. This is confirmed by the experimental results as well where SAGD seems to outperform other techniques when the dataset size is smaller. I found the bounds on the convergence to a population stationary point quite interesting as well. What is curious is that adding the laplacian noise to the minibatch gradient seems to fix the convergence issues of minibatch RMSProp that were raised by Reddi et al in “On the Convergence of Adam and Beyond”?

Weaknesses: The main weakness of this paper is the experimental section. I would have liked to see more thorough and rigorous experiments. For example, how is SAGM affected by the size of the minibatches in these experiments? I am also not sure if the baselines were tuned properly. This is because both ResNet18 and VGG19 should be reaching slightly higher test accuracies with SGD/Adam than those reported in the paper. My other question is: does RMSProp offer any particular advantage to being used with DPG-LAG/DPG-SPARSE? The method proposed seems general enough to be used with any first order optimization algorithm? If that is indeed the case, then it would have been good to also include experiments on how DPG-LAG/DPG-SPARSE might affect generalization of other popular algorithms like SGD/Adam.

Correctness: Yes.

Clarity: Overall the paper is well written and is easy to follow. My only comment about the writing is that, while it is true that typically by just tuning the learning rate, adaptive gradient methods might generalize worse than SGD, it is worth mentioning that if all the hyperparameters of Adam are tuned properly, it can often reach the generalization performance of SGD. This was shown by Choi et al in “On Empirical Comparisons of Optimizers for Deep Learning”.

Relation to Prior Work: See the answer to the above question. I am not familiar with related work in this area in the differential privacy or adaptive data analysis literature, so I cannot be sure that all related work in those areas were discussed.

Reproducibility: Yes

Additional Feedback: A couple of additional questions: 1. How do the high probability bounds change when using mini-batches of size m? 2. Is data augmentation used in the experiments? Data augmentation is a popular technique to improve the generalization of neural network models, and It could be thought of as extending the dataset size, albeit in a more biased way. It would be interesting to compare the performance of DPG-LAG/DPG-SPARSE with simple data augmentation techniques.


Review 3

Summary and Contributions: The authors propose Stable Adaptive Gradient Descent (SAGD) which utilizes differential privacy based approaches to guarantee better generalization for adaptive gradient methods. The authors provide both theoretical analyses as well as comprehensive experiments to demonstrate the improvements in performance attained by SAGD. Post Rebuttal: The authors have addressed some of my concerns but the lack of code and other reviews imply that I keep my current score.

Strengths: The paper adopts the simple idea of adding a Laplacian noise to a gradient estimate. This idea is inspired by differential privacy and provides a simple practical variant of adaptive methods. The paper provides convergence results to first-order stationary points for the full gradient version, the DPG-Sparse analog as well as the mini-batch version. The main strength of this paper lies in the applicability of their approach to most existing popular adaptive methods. The experimental section is quite strong: the authors evaluate their results over many runs, calculate the mean and standard deviation while reporting results, evaluate the performance for the different datasets and perform extensive hyperparameter tuning.

Weaknesses: 1. It is unclear how guaranteeing stationary points that have small gradient norms translates to good generalization. The bounds just indicate that these algorithms reach one of the many stationary points for adaptive gradient methods and don't talk about how reaching one of the potentially many population stationary points especially in the non-convex regime can translate to good generalization. A remark on this would be helpful. 2. Line 124-125: For any w, the Hoeffding's bound holds true as long as the samples are drawn independently and so it is always possible to show inequality (2). Stochastic algorithms moreover impose conditioning on the previous iterate further guaranteeing that Hoeffding inequality holds. It will be great if the authors can elaborate on this. 3. The bounds in Theorem 1 have a dependence on d, which the authors have discussed. However, if \mu is small, the bounds are moot. If \mu is large, then the concentration guarantees are not very useful. Based on values in Theorem 2, latter seems to be the case. 4. It seems weird that the bounds in Theorems 2 and 4 do not depend on the initialization w_0 but on w_1. 5. For experiments on Penn-Tree bank, it seems that the algorithms are not stable with respect to train perplexity.

Correctness: There are some concerns about the claims made in this paper (see point 2 in Weaknesses). The empirical methodology is very sound and the authors provide the details for hyper-parameter tuning, applicability to different datasets, and comparison against a comprehensive set of algorithms. (A github link to the implementation will be

Clarity: The paper is well written and easy to follow.

Relation to Prior Work: This needs improvement as the authors miss out on a lot of theoretical progress made in the field of adaptive gradient methods.

Reproducibility: No

Additional Feedback: One disadvantage is that authors have not shared the code with respect to reproducibility. This has been an increasing concern in optimization papers regarding this a link to the code will make the case for this paper strong. Typo in Line 134


Review 4

Summary and Contributions: *** POST AUTHOR FEEDBACK *** After reading the rebuttal, I still have some concerns about the experiments. I strongly recommend the authors to include a new experiment (see <2>) similar to [Wilson et al., 2017] since this new experiment is more useful than the experiments given in the paper. <1> The SGD with Gaussian noise seems to be a straightforward baseline, which is missing in the experiments. Missing this baseline method makes me wonder why DP is useful since the focus of this paper is on generalization. Ideally, the authors should give a bound for this baseline method from the DP perspective and improve this baseline method using DP (e.g., use Laplace noise instead of Gaussian noise). In the rebuttal, the authors claim that "We believe it is of great interest to show how Gaussian noise works in our setting." and "We consider the theoretical details and experimental results as a future work." The response could suggest that SGD+Gaussian noise is as good as the proposed method. Moreover, the effectiveness of SGD+Gaussian noise can be explained from a non-DP perspective, which makes me wonder why DP is useful in this setting. <2> The experimental setup is non-standard, which is another concern. In the rebuttal, the authors claim that "[Wilson et al., 2017] mainly plotted the training/test accuracy against the number of epochs. We agree that it would be interesting to add experiments to compare SGD with differential privacy." However, it seems that the authors will not add a new experiment to show that SGD+Laplace noise gives a better generalization error compared to SGD or SGD+Gaussian noise in the standard setup. If the proposed method works well, it should be very easy to reproduce the results of [Wilson et al., 2017] alongside the proposed method. I think in practice this new experiment is more useful than the experiments given in the paper. ------------ In this work, the authors propose using differential privacy to generate noisy gradients and show that such a procedure can lead to better generalization.

Strengths: The authors show that using differential privacy can lead to better generalization. Furthermore, the authors show that differential privacy preserves the statistical nature of gradients with provable guarantees. Finally, the authors empirically demonstrate that an adaptive method built on differential privacy could lead to better test results. 

Weaknesses: The empirical expereiment design mainly follows Wilson et al 2017. However, the authors use different evalution metrics, which makes it incomparable to the results in Wilson et al 2017. It will be great if the authors reproduce most of results in Wilson et al 2017 alongside the proposed method using the same evaluation metrics for a fair comparison. The empirical evaluation is not very thorough. (see the additional feedback)

Correctness: The proposed method seems to be solid. I do not check the theorical results since I am not familiar with differential privacy

Clarity: This paper is well written.

Relation to Prior Work: I am not aware of prior works using differential privacy to achieve better generalization.

Reproducibility: Yes

Additional Feedback: I have some questions. Does SGD using gradients generated from differential privacy (e.g., Laplace noise) achieve the same generalization as the proposed method? In other words, does the method perform well if I set v_t to be 0 (or set \beta_2 to be 1) in Algorithm 1-3. A follow-up question is whether SGD with gradient corrupted by Gaussian noise performs well or not. If so, why not use Gaussian noise since in this setting the focus is on generalization instead of differential privacy? I think SGD with Gaussian noise is a straightforward baseline method. The behavior of this baseline method is also well-studied. For example, see Mandt et al 2016 [A Variational Analysis of Stochastic Gradient Algorithms]. The last question is whether the proposed method works well for small datasets in terms of generalization?

[Author Response · NeurIPS 2020]

We thank all the reviewers for their insightful feedback which do help us improve the quality of our paper. We explain how we address your concerns and revise our paper based on your comments. Based on **R1** and **R3** concerns about the code, we are more than happy to share it. If they wish to see the code right away, we can share the code through AC/PC.

**Reviewer 1:**

– *"There are relatively few stable facts. This paper does not necessarily reduce the entropy."* The reviewer raised a very important point. We agree that there are tremendous amount of papers on this topic with few stable facts. We expect our work to bring new insights to this field, especially in understanding the generalization via the lens of differential privacy.

– *" Figure 2"* We thank the reviewer for pointing this out. We will improve the quality of the plots in the revision.

– *"broader impact"* This paragraph will be improved to reflect the idea of avoiding over-fitting (see plot (b) Figure 2).

**Reviewer 2:**

– *"I would have liked to see more thorough and rigorous experiments."* We mainly follow the method in [Wilson et al., 2017] to tune the step size, since they highlight that the initial step size and the scheme of decaying have a considerable impact. We agree with the reviewer that the mini-batch size and hyper-parameter tuning would also play an important role in the performance. Still, we think that our experiments provide an extensive experimental evaluation of variants of training algorithms for various tasks such as image classification and language modeling. We believe our experiments offer a fair comparison since the same effort was done to tune the hyper-parameters for each baseline.

– *"Does RMSProp offer any particular advantage..."* We agree that DPG-LAG/DPG-SPARSE can be used with any first order optimization algorithm. The RMSProp can be viewed as SGD when $\beta_2 = 1$. We plan to provide a generic stable adaptive algorithm that encapsulates many popular adaptive and *non-adaptive* methods in the Appendix.

– *"How do the high probability bounds change when using mini-batches of size m?"* The high probability bounds on the gradient mainly follow the generalization guarantee of differential privacy with conditions on the privacy parameters $(\epsilon, \delta)$ and sample complexity. In the case of mini-batch, the value of privacy parameters $(\epsilon, \delta)$ and the condition on sample complexity get modified. We have provided details in the proof of Theorem 5, see Section B.2 of the Appendix.

– *"Is data augmentation used in the experiments?"* We used data augmentation for MNIST and CIFAR-10. For MNIST, we normalize the value of each feature to [0,1]. For CIFAR-10, we normalize, randomly crop and rotate the images.

**Reviewer 3:**

– *"It is unclear how guaranteeing stationary points that have small gradient norms translates to good generalization"* Our main theoretical results provide the convergence to the *'population stationary point'*. Note that Theorems 2, 4 and 5 show the convergence of the norm of the *population gradient* instead of the empirical gradient. Also, one will be able to use our results to establish the generalization error of the loss function based on arguments such as the PL condition.

– *"The Hoeffding's bound holds true as long as the samples are drawn independently"*. Yes, Hoeffding's bound holds as long as the samples are drawn independently. However, in the setting of *sample reuse* (setting in this paper) such as SGD with multi-pass, the reused samples are not independent anymore, since the posterior distributions of samples change after training on the reused samples.

– *"The bounds in Theorem 1 have a dependence on d"*. The reviewer raised a very interesting question! Yes, the dependence on $d$ is a known result for differential privacy (DP) and is hard to avoid (see ref. [1]). Some works on DP try to improve this dependence on $d$ by leveraging special structures of the gradients. This will be considered in the future.

– *"do not depend on the initialization $\mathbf{w}_0$ but on $\mathbf{w}_1$."* We thank the reviewer for this typo: should be $\mathbf{w}_0$ instead of $\mathbf{w}_1$.

– *"For Penn-Tree bank,[...] algorithms are not stable w.r.t. train perplexity."* With respect to train perplexity, all methods stabilize around a target value (which is of course different given the highly nonconvex loss). We note that the test perplexity increases after several epochs for most baselines while our method keeps a low and steady one.

**Reviewer 4:**

– *"experiment design mainly follows [Wilson et al., 2017]"* The design is different from [Wilson et al., 2017] (except for the stepsize tuning, see **Reviewer 2**). Indeed, we study the *generalization* performance of each algorithm with an *increasing* training sample size $n$ (see Fig. 1, x-axis is $n$). This is consistent with our theoretical results which show the convergence of SAGD in terms of $n$. However, [Wilson et al., 2017] mainly plotted the training/test accuracy against the number of epochs. We agree that it would be interesting to add experiments to compare SGD with differential privacy.

– *"SGD with gradient corrupted by Gaussian noise performs well or not"* Excellent question and nice reference! Actually, one can also use Gaussian noise to design a differentially private algorithm (namely Gaussian Mechanism [7]). Also, there are papers showing the connection between SGLD (Stochastic Gradient Langevin Dynamics) and differential privacy. Yet, the existing generalization bound of SGLD is established by the techniques of algorithmic stability [23, 26], which scales with $(\sqrt{T})$. We believe it is of great interest to show how Gaussian noise works in our setting. We will add a discussion in the paper. We consider the theoretical details and experimental results as a future work.

– *"whether the proposed method works well for small datasets in terms of generalization"* Figure 1 shows that SAGD has a slightly better test accuracy than other algorithms when the training sample size $n$ is small (x-axis).

[Meta-Review · NeurIPS 2020]

While the central idea and detailed experimentation were appreciated unanimously by the reviewers and therefore I am recommending accept, there are multiple issues paper that the authors are enouraged to address, including a comparison with the SGD+DP baseline - even for the theoretical considerations. Furthermore the experiments presented by the paper are not run to completion (100 epochs) and therefore they do not acieve SOTA numbers - this should be fixed.